# SARS-CoV-2 variants of concern in children and adolescents with COVID-19: a systematic review

Margarethe Wiedenmann ![ORCID],[1,2] Aziz Mert Ipekci,[3] Lucia Araujo-Chaveron ![ORCID],[4,5] Nirmala Prajapati,[6,7] Yin Ting Lam ![ORCID],[3] Muhammad Irfanul Alam ![ORCID],[8] Arnaud G L'Huillier,[9] Ivan Zhelyazkov,[10] Leonie Heron,[3] Nicola Low ![ORCID],[3] Myrofora Goutaki ![ORCID] [3]

NL and MG contributed equally.

For numbered affiliations see end of article.

**Correspondence to**
Dr Margarethe Wiedenmann;
margarethe.wiedenmann@swisstph.ch

## ABSTRACT

**Objectives** Infections by SARS-CoV-2 variants of concern (VOCs) might affect children and adolescents differently than earlier viral lineages. We aimed to address five questions about SARS-CoV-2 VOC infections in children and adolescents: (1) symptoms and severity, (2) risk factors for severe disease, (3) the risk of infection, (4) the risk of transmission and (5) long-term consequences following a VOC infection.

**Design** Systematic review.

**Data sources** The COVID-19 Open Access Project database was searched up to 1 March 2022 and PubMed was searched up to 9 May 2022.

**Eligibility criteria** We included observational studies about Alpha, Beta, Gamma, Delta and Omicron VOCs among ≤18-year-olds. We included studies in English, German, French, Greek, Italian, Spanish and Turkish.

**Data extraction and synthesis** Two reviewers extracted and verified the data and assessed the risk of bias. We descriptively synthesised the data and assessed the risks of bias at the outcome level.

**Results** We included 53 articles. Most children with any VOC infection presented with mild disease, with more severe disease being described with the Delta or the Gamma VOC. Diabetes and obesity were reported as risk factors for severe disease during the whole pandemic period. The risk of becoming infected with a SARS-CoV-2 VOC seemed to increase with age, while in daycare settings the risk of onward transmission of VOCs was higher for younger than older children or partially vaccinated adults. Long-term symptoms following an infection with a VOC were described in <5% of children and adolescents.

**Conclusion** Overall patterns of SARS-CoV-2 VOC infections in children and adolescents are similar to those of earlier lineages. Comparisons between different pandemic periods, countries and age groups should be improved with complete reporting of relevant contextual factors, including VOCs, vaccination status of study participants and the risk of exposure of the population to SARS-CoV-2.

**PROSPERO registration number** CRD42022295207.

## STRENGTHS AND LIMITATIONS OF THIS STUDY

⇒ This review is based on a wide range of worldwide studies found during repeated systematic searches of different databases, reducing our risk of having missed relevant information.
⇒ We included studies in the various languages the review team spoke, only excluding three studies for language reasons.
⇒ We assessed the included studies' risk of bias at the outcome level to evaluate the quality of the studies included.
⇒ The included studies did not provide information on the specific Omicron subvariants, limiting the interpretation of the respective data.

## INTRODUCTION

The role of children and adolescents in the pandemic of severe acute respiratory syndrome coronavirus 2 (SARS-CoV-2), the cause of coronavirus disease 2019 (COVID-19), remains an important topic of discussion. In our systematic review, covering the period when SARS-CoV-2 ancestral strains circulated, we found that children (≤12 years of age) and adolescents (13–17 years of age) usually present with mild disease, with a minority of children of all ages (<18 years) remaining asymptomatic throughout the course of infection and their risk of acquiring and transmitting SARS-CoV-2 seemingly increases with age.[1]

Since late 2020 SARS-CoV-2 variants of concern (VOCs) have emerged, with the Alpha variant (previously referred to as B.1.1.7) being the first. These lineages possess one or more mutations, which are associated with different phenotypic properties, such as changes in the severity of disease, increased transmissibility or the ability to evade infection-induced or vaccine-induced immunity.[2] The circulation of Alpha, Beta, Gamma, Delta variants and the Omicron group of variants and subvariants has led to large waves of COVID-19 in different countries.[3] At around the same time as the first VOCs emerged and spread, vaccines that gave protection against infection and severe

disease caused by the original SARS-CoV-2 lineage were introduced in adults.[4][5] Vaccines were first licensed for 16–18 year-olds in spring 2021 and later for children <16 years old, but vaccination availability and uptake varied by country.[6] Also, preventive interventions introduced at the population level have been relaxed in many countries. A re-evaluation of the role of children and adolescents in the SARS-CoV-2 pandemic is warranted, given the potential for the mutations in VOCs to affect age-associated patterns of COVID-19 clinical disease and SARS-CoV-2 transmissibility and the higher numbers of children and adolescents testing positive for SARS-CoV-2 than in the pre-VOC period.

The objectives of this study were to address five research topics about SARS-CoV-2 VOC infections in children and adolescents younger than 20 years: (1) What are the symptoms caused by SARS-CoV-2 VOCs in children and adolescents, and do they differ from those of SARS-CoV-2 wild type infections? (2) What are the risk factors for severe COVID-19 in children and adolescents infected with a SARS-CoV-2 VOC? (3) Does the risk of infection with a SARS-CoV-2 VOC differ between children, adolescents and adults who have not received a COVID-19 vaccine? (4) Does the risk of transmission of a SARS-CoV-2 VOC differ between children, adolescents and adults who have not received a COVID-19 vaccine? (5) Does the risk of developing long-term effects (duration >28 days) after infection with a SARS-CoV-2 VOC differ between children, adolescents and adults who have not received a COVID-19 vaccine?

## METHODS
### Design
We registered a protocol for this review in the International Prospective Register of Systematic Reviews (PROSPERO).[7] We report the study according to the Preferred Reporting Items for Systematic Review and Meta-Analyses guidelines (online supplemental table 1).[8]

### Search strategy
We searched: (1) the COVID-19 Open Access Project database, which aggregates COVID-19 research from PubMed, EMBASE, PsycINFO, bioRxiv and medRxiv from 1 January 2020 to 1 March 2022 and (2) PubMed using Medical Subject Headings (MeSH) terms from 1 August 2020 to 7 March 2022.[9] Owing to large numbers of records that did not fulfil the inclusion criteria, we revised the PubMed search strategy, including each VOC, identified by their Greek letter and pangolin lineage terminology and conducted a new search between 8 March and 9 May 2022. We also searched reference lists of systematic reviews and included studies, and we included studies suggested by experts in the field, but not identified by our search. Full search terms are included in the online supplemental note 1.

### Eligibility and screening
Observational epidemiological studies (case series with more than one case, cross-sectional studies, case–control studies and cohort studies) and modelling studies that were based on the analysis of the original data were eligible for inclusion if data collection occurred during a period when VOCs were predominant in the study setting. The presence of a VOC could be based on genetic sequencing in the study or by assumptions based on the authors' reporting, the dominant lineage at the time of testing and the study location. We did not apply any geographical or language restrictions to the search, but we only read the full text of papers written in languages spoken fluently by the review team (English, German, French, Greek, Italian, Spanish or Turkish). Identified titles and abstracts were screened for potential eligibility by one member from an international team of 10 reviewers, including records about which they were unsure. A second member verified the selection. If either reviewer was unsure about the potential eligibility, we obtained the full-text. We followed the same approach for the full-text assessment. A third reviewer resolved disagreements.

### Data extraction
We developed, piloted and revised a data extraction template in Microsoft Excel at the beginning of the study. One reviewer extracted data from each included study and a second reviewer verified the extraction. Extracted data included information on the publication characteristics, study design, study setting, study population overall and those younger than 20 years old, age groups, prevalent VOC(s) at the time of study and outcomes for each relevant review question.

### Risk of bias assessment
We assessed the risk of bias at the outcome level using published tools for cohort and case–control studies.[10][11] The credibility and relevance of modelling studies was evaluated using a published checklist.[12] To help with the interpretation of the findings, we also recorded whether the authors of included studies reported on the stage of the SARS-CoV-2 pandemic, on any existing control measures or on the vaccination status of the study participants at the time of the study.

### Data synthesis
We synthesised the information from the included studies descriptively for each question. We tabulated summaries of the data extracted, including the number of studies identified for each VOC, the age groups studied and the outcomes reported.

Definitions of disease severity differed between studies. To allow comparisons, we applied the following categorisation: mild or asymptomatic disease (patients were not hospitalised), moderate disease (patients were hospitalised, but did not require intensive care), severe disease (patients were hospitalised and intensive care was necessary). If we did not have sufficient information to apply these categories, we used the authors' definitions and reported these in the text. Where possible, we presented the results separately for different age groups: neonates

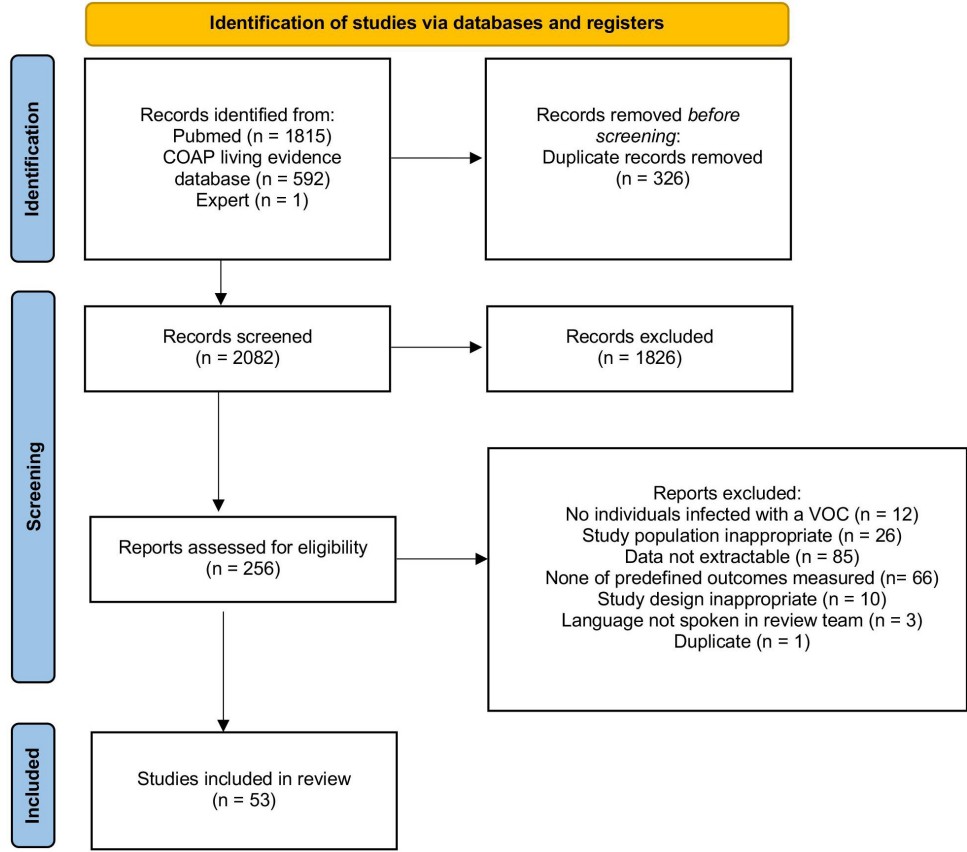

**Figure 1** Flow chart. Of the 2407 studies found via database searches 53 were found to be eligible for this systematic review (including one study recommended to us by experts). From Page *et al*.[76] COAP, COVID-19 Open Access Project; VOC, variants of concern.

and infants 0–1 years old; young children 1–4 years old; older children 5–9 years old; younger and older adolescents 10–20 years old. If the intended age groups were not reported, we used the groups reported by study authors. We categorised the country in which the study was conducted according to World Bank income groups as a simplified proxy for the level of government resources spent on health services and public health surveillance activities.[13] We report our findings descriptively. There were too few studies that reported quantitative findings in a comparable way to conduct a meta-analysis. Any comparison with non-VOC lineages (any strain not considered by the WHO as a variant of concern) included in this manuscript were made by the authors of the original manuscripts.[2] We did not make independent comparisons between outcomes during VOC and non-VOC periods.

**Patient and public involvement**
None.

**RESULTS**
We screened 2407 articles published up to 9 May 2022, 53 of which were included (figure 1 and online supplemental figures 1–4). According to World Bank data, 47 (89%) articles were from 14 high-income countries, 6 (11%) were from 5 upper middle-income countries and

none was from a lower middle-income or low-income country (table 1 and figure 2).

Of the 53 studies included in the review, 44 (83%) were cohort studies, 4 (7.5%) were cross-sectional studies, 3 (5.7%) were case–control studies, 1 (1.9%) was a modelling study and 1 (1.9%) was a case series (table 1). Some included studies compared different variants and others different age groups, infected with the same variant.

Most studies were judged to be at moderate risk of bias in most domains. There were high risks of bias in a few domains especially regarding matching of exposed and unexposed cohorts as well as the assessment of prognostic factors (irrespective of the research question they addressed) as most studies did not control for possible confounding factors (online supplemental tables 2–6).

**Review question 1: symptoms and severity of disease among children and adolescents infected with VOCs**
**Symptoms**
Symptoms experienced by children and adolescents infected with a VOC were presented in 19 studies involving the Alpha (n=7), Delta (n=6), Omicron (n=4) or Gamma variant (n=1) (online supplemental tables 7–10).[14–32] Children and adolescents experienced a wide range of symptoms. We did not identify any symptoms consistently associated with a specific VOC or whether symptom

**Table 1** Characteristics of included studies

| Characteristic | Total | Variant of concern | | | | |
| --- | --- | --- | --- | --- | --- | --- |
| | | Alpha | Beta | Gamma | Delta | Omicron |
| Number of studies, n* | 53 | 27 | 2 | 2 | 27 | 12 |
| Review question addressed, n* | | | | | | |
| Review question no. 1 | 35 | 13 | 2 | 2 | 21 | 11 |
| Review question no. 2 | 2 | 1 | 1 | 1 | 2 | 2 |
| Review question no. 3 | 15 | 10 | – | – | 4 | 1 |
| Review question no. 4 | 15 | 13 | – | – | 3 | 1 |
| Review question no. 5 | – | 2 | 1 | 1 | 2 | – |
| Study design | | | | | | |
| Cohort | 44 | 21 | 2 | 2 | 24 | 9 |
| Case–control | 3 | 3 | – | – | – | – |
| Cross-sectional | 4 | 2 | – | – | 3 | 2 |
| Modelling | 1 | 1 | – | – | – | – |
| Case series | 1 | – | – | – | – | 1 |
| Income category according to World Bank categories, n | | | | | | |
| High income | 47 | 26 | 1 | 1 | 23 | 11 |
| Upper middle income | 6 | 1 | 1 | 1 | 4 | 1 |
| Lower middle income | – | – | – | – | – | – |
| Low income | – | – | – | – | – | – |

The added number for each VOC is more than 53 as some studies reported on more than just one VOC.
*The total number of studies included is 53.
VOCs, variants of concern.

frequencies with VOC infections differed from the pre-VOC period. One study reported on croup in children and adolescents infected with the Omicron variant and found an increase in croup cases in hospital inpatients and outpatients compared with the pre-Omicron period (median weekly croup cases (Omicron vs pre-Omicron): 11 (IQR 2–17) vs 0 (IQR 0–0), p<0.001) (online supplemental table 9).[14] We did not find any studies investigating symptoms in children infected with the Beta variant. A detailed comparison between studies was not possible owing to differences in study populations (community or healthcare setting), symptoms recorded and age groups reported.[15–32]

### Severity

Children and adolescents mostly experienced mild disease when infected with a SARS-CoV-2 VOC. More severe infections were described with the Gamma variant compared with pre-VOC variants. On the other hand, infections with the Omicron variant were less severe compared with Delta variant infections.

Paediatric disease severity associated with SARS-CoV-2 VOC infections was described in 25 studies (online supplemental tables 7–10).[18–20 26–29 31–47] The Alpha variant did not appear to result in more severe COVID-19 disease than disease caused by non-VOCs in children and adolescents in five studies.[18–20 33 34] In two studies reporting on paediatric inpatients and outpatients one from Brazil and one from the USA, the overall hospitalisation rate as well as the intensive care unit (ICU) admission rate of children infected with the Gamma variant was higher than for non-VOCs (Gamma vs non-VOC: 28.3% vs 24.9%,[35] Gamma vs non-VOC: 5.2% vs 1.2%[28]). Severity of Delta variant infection was compared with the pre-Delta period (including periods where previous VOCs where predominant) in eight studies but the reported data were inconsistent, and differences in disease severity could not be assessed.[26–28 36–40] Studies of Omicron variant infection described less severe disease than infection with the Delta variant in children and adolescents <18 years old, based on data from nine studies from South Africa, Qatar and the USA.[29 31 32 41–46]

One study from Denmark analysed the symptoms and severity of disease experienced by children and adolescents younger than 17 years old, hospitalised for multi-inflammatory syndrome in children during the Delta and non-VOC predominant periods. The range of symptoms was similar in both periods, as was the proportion of children admitted to ICU (Delta, 55% (28/51) vs non-VOC 52% (12/23)).[47]

We did not find any studies showing an association between the SARS-CoV-2 Omicron lineages and the global outbreak of paediatric hepatitis of unknown origin, which has been attributed to adenoviruses in spring 2022.[48]

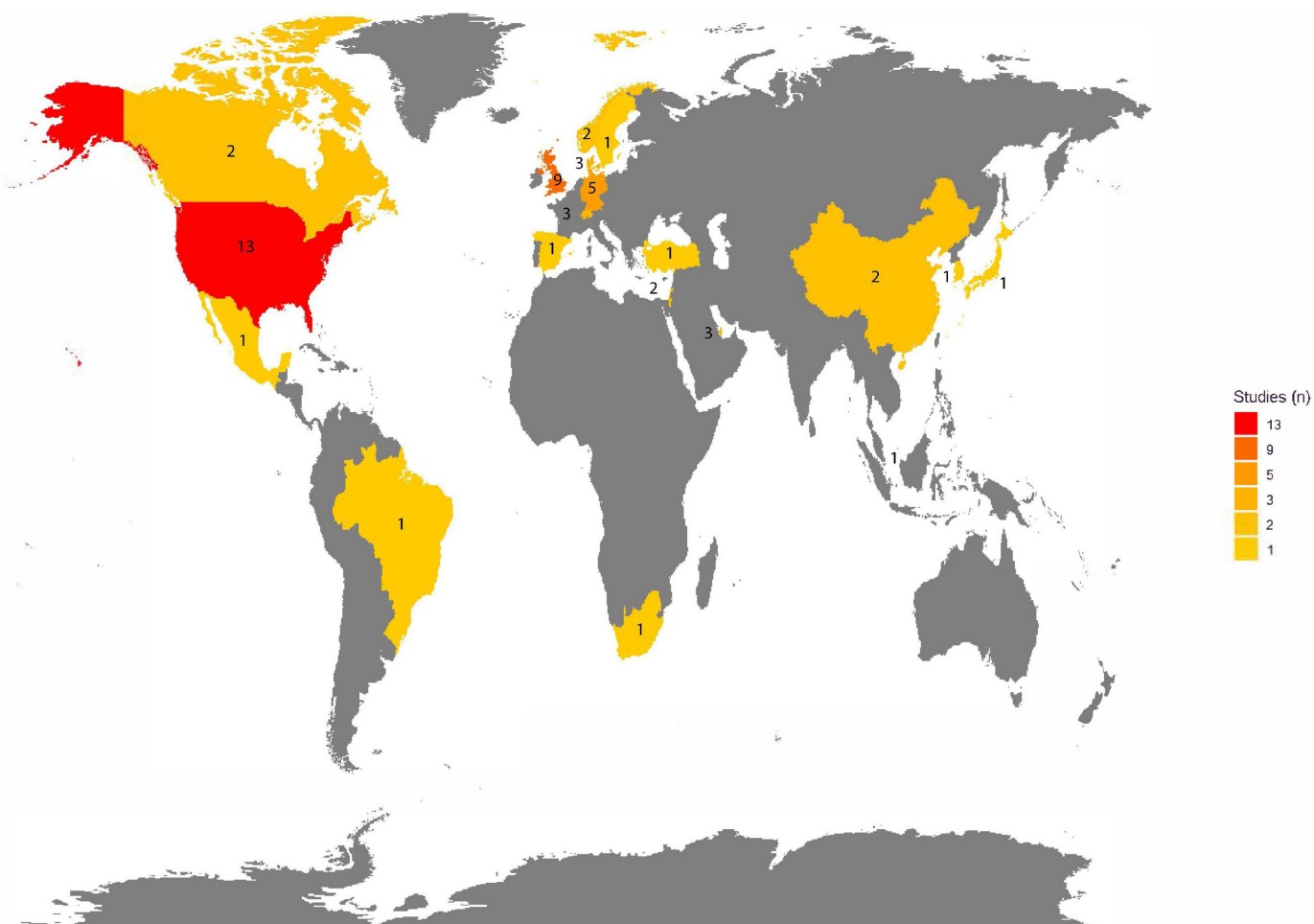

**Figure 2** Map of the number of studies conducted in each country. Most studies came from high-income countries (n=47, 89%). Six studies (11%) were from upper middle-income countries.

No studies reported on how symptoms or severity differed between different paediatric age groups or compared children and adolescents infected with a VOC to adults infected with the same variant.

### Review question 2: risk factors for severe COVID-19 in children and adolescents infected with a SARS-CoV-2 VOC

We found one article that used data from the National COVID-19 database in Qatar and compared risk factors for severe COVID-19 in children and adolescents infected with the Omicron and Delta variants. The study showed that children and adolescents <18 years old with ≥1 comorbidity (mostly chronic lung disease) were more likely to have moderate or severe disease than those without comorbidities if they were infected with the Omicron lineage (adjusted OR (aOR) 3.2, 95% CI 1.1 to 9.0), but not if they were infected with the Delta lineage (aOR 0.8, 95% CI 0.5 to 1.4).[31] One paper reported on risk factors for severe COVID-19 disease in children 5–11 years old during the whole pandemic period, including the time when the Omicron variant became predominant. The study assessed several possible risk factors and found evidence of an increased risk of severe disease in children with diabetes or obesity.[45]

### Review question 3: risk of becoming infected with a SARS-CoV-2 VOC

The risk of becoming infected with a SARS-CoV-2 VOC was analysed in time periods when the Alpha, Delta or Omicron variants were predominant in 15 studies (online supplemental table 11).[15 17–19 33 49–58] Immunity due to previous infection or vaccination appeared to be a relevant factor in determining the risk of infection but these factors were poorly reported in the included studies. Some studies (4/15) summarised the vaccination status of the entire study sample, which ranged from an undefined proportion of partially vaccinated individuals to 70% fully vaccinated, without analysing it at the individual level for adults and children and adolescents.[18 49–51]

Six studies evaluated data from Alpha variant outbreaks in daycare centres and compared the risk of infection between children and adults.[15 19 52–55] They generally showed a lower attack rate for infection with the Alpha variant ranging from 0% to 27% for young children (mostly ≤6 years old) than for adults (0%–71%). However, there were large variations in attack rates between different facilities or classes at the same facility. The study

participants' vaccination status was not reported in any of these studies.

One study reported the attack rates of children by age group. The attack rates for young children (≤4 years old) decreased with increasing age from 44% to 4%, with the highest rate for infants <1 year old and the lowest for children 3–4 years old.[15] The attack rates of older children (10–19 years old) and adults (30–39 years old) were similar in one study from Qatar, but the vaccination status was not reported.[56]

In one study analysing household transmission of the Alpha variant in the USA, the risk of infection was high in both younger children ≤11 years old (78%, 95% CI 51% to 76%) and older children and adolescents 12–17 years old (56%, 95% CI 40% to 72%) with overlapping CIs.[18] Compared with the pre-VOC period, a similar or lower risk of infection was reported in three household or school studies for children and adolescents of all age groups when the Alpha variant was predominant.[18 19 52] However, in two studies from Israel, including a large hospital study, a higher risk of infection for the Alpha variant than for non-VOCs was described for all paediatric age groups.[17 33] Data on the risk of infection during the Delta predominant time period analysed in four studies comparing children and adolescents to adults were inconclusive.[49–51 57] Two of these four studies, one cohort study from Singapore and one from the UK found the risk of infection to be higher for younger children ≤11 years old/school-aged children 5–18 years old than for adults who were either completely or partially vaccinated.[50 51] One study analysed the risk of infection in daycare centres during the Omicron predominant time period and found it to be slightly lower for young children 3–7 years old (44%) than for adults (50%). As described, during the Alpha-predominant period large variations in attack rates (15.4%–61.1%) were observed between different school classes.[58]

### Review question 4: risk of SARS-CoV-2 transmission from children and adolescent infected with a VOC

The risk of SARS-CoV-2 transmission was evaluated in time periods when Alpha, Delta or Omicron variants were predominant in 14 studies (online supplemental table 12).[18 33 49 54 58–67]

Overall, the transmission risk with a VOC was reported to be higher for younger children in most studies. A higher risk of onward SARS-CoV-2 Alpha variant transmission in household settings from younger children <10 years old than from older children 10–19 years old and adults was found in four studies.[54 59–61] These studies did not report on whether the participating adults or older children were vaccinated. One household study from the USA found comparable transmission of Alpha variant infections from children ≤11 years old compared with adolescents 12–17 years old (97% of whom were unvaccinated) (OR 1.37, 95% CI 0.61 to 3.04).[18] The risk of transmission from children <10 years old with an Alpha variant infection was found to be higher than from

children with a pre-VOC or non-VOC infection in three studies.[33 61 62] One study found the transmission risk from 10 to 19 year-olds with an Alpha variant infection to be similar to that of pre-VOC infection.[62] Comparing the risk of onward transmission between age groups for participants infected with either the Alpha or Delta variant, the risk was similar for younger children <12 years old and lower for older children and adolescent 10–19 years old, compared with 30–39 year old adults.[63 64] One household transmission study from Singapore found the transmission risk of younger children ≤11 years old infected with the Delta variant to be higher (25%) than the transmission risk of young adults 18–29 years old who were at least partially vaccinated (9%).[49] The availability of evidence on the risk of onward transmission of children with an Omicron variant infection was limited. We only found one small study that analysed the risk of onward transmission of young children ≤6 years old with an Omicron variant infection in a daycare setting and found it to be lower (3/24, 12.5%) than the risk of onward transmission from adults (1/2, 50%).[58]

### Review question 5: risk of long-term symptoms or signs (duration >28 days) following an infection with a SARS-CoV-2 VOC

Few children reported long-term symptoms or signs of infection with a defined duration of >28 days.[26 27] The risk for children and adolescents experiencing symptoms longer than 28 days was similar for children infected with the Alpha (12/694, 1.73%) and the Delta lineage (15/706, 2.12%) in the UK, and was similar for younger (5–11 years old) compared with older (12–17 years old) children infected with the same variant. For those children with longer illness duration, no new symptoms developed after day 28.[26] A study comparing Danish children who tested positive for SARS-CoV-2 with a reference cohort of children who tested negative during the overall pandemic period (Alpha and Delta variant predominant times included) found that 0.12% (95% CI 0.09% to 0.15%) of children experienced symptoms for more than 29 days (58 of 48 948 children).[27]

### DISCUSSION

Children and adolescents (<20 years old) with a SARS-CoV-2 VOC infection mostly experienced mild disease and the symptoms experienced did not vary widely between VOCs. Compared with non-VOC and other VOC induced infections more severe infection was observed with either the Delta or the Gamma SARS-CoV-2 variant. Compared with otherwise healthy peers, children and adolescents with comorbidities appeared to have an increased risk of infection by Omicron, but not Delta, and diabetes and obesity were reported as risk factors for severe infection during the whole pandemic period. The studies reviewed suggest that the risk of SARS-CoV-2 infection in daycare centres was lower overall for younger children than for adults, while the risk compared with adults decreased

with increasing age of the children. The transmission risk of SARS-CoV-2 from younger children infected with a VOC seemed to have been higher than that of older children and adults as well. The proportion of children and adolescents reporting symptoms more than 28 days after a SARS-CoV-2 VOC infection was low. Interpretation and generalisation of the evidence was limited by inconsistent reporting of age groups and by inadequate reporting of vaccination and other control measures in the included studies.

## Strength and limitations

We used a systematic approach to search different databases, covering the period of VOC circulation that includes the first Omicron variants (BA.1 and BA.2) in 2022. Unlike preceding VOCs, Omicron comprises an increasingly large groups of multiple variants and subvariants. Information on the specific subvariants was not provided in the included studies, which limits the interpretation of the associated data in our review. We included a broad selection of study designs in various languages and screened reference lists of included studies. Our search strategy might have missed relevant studies, although we only excluded three records for language reasons. A limitation of the review is that misclassification of the variant could have occurred as only a minority of study samples from all cases was sequenced in several studies and we relied on dates during which a variant was dominant. Non-differential misclassification would reduce the magnitude of associations with VOC infection status.

## Interpretation and implications of findings

The severity of SARS-CoV-2 infections, the risk of becoming infected and the risk of onward transmission depend on both intrinsic factors associated with the person (eg, host immunity) and the pathogen (eg, variant/subvariant transmissibility, virulence, immune evasion) and extrinsic factors (eg, study setting, COVID-19 at the population level, vaccination status of study participants, preventive measures in place). Differentiation between the contributions of intrinsic and extrinsic factors in studies included in this review was challenging because frequent deficiencies in reporting of extrinsic factors hindered comparisons and interpretation of outcomes from different regions or different pandemic periods, even from the same country. All studies were conducted in high or upper middle-income countries, which might affect the generalisability of the review findings to children in settings with less-resourced health systems, lower vaccine coverage and different patterns of comorbidity and co-infection with other respiratory viruses. We assessed the risk of bias using published tools for cohort and case–control studies but found that the selection bias was hard to assess, especially for studies addressing the symptoms and severity of disease in hospitalised populations, since reporting of

indications for hospitalisation (especially for children) varied between countries and between different stages of the pandemic.[10 11]

## Risk of exposure to, and transmission of, SARS-CoV-2

The risk of exposure to SARS-CoV-2 has a direct effect on the risk of becoming infected, which influences the risk of onward transmission. Younger children are more dependent on adult care takers, who were most likely to be the index case, than older children and adolescents. This increased risk of exposure to infected adults might explain their higher risk of becoming infected and for onward transmission compared with older children and adolescents. The risk of exposure to SARS-CoV-2 also influences the interpretation of the data on disease severity in children and adolescents with chronic diseases. We found an increased risk of severe disease for children with diabetes or obesity, but not for children and adolescents with respiratory conditions, except if they were infected with Omicron.[45] As high-risk populations have been known to take additional measures to stay safe since the beginning of the pandemic, these findings could be partly explained by more intense shielding.[68]

## Vaccination status

Vaccines protecting people from severe COVID-19 were first licensed for adults at the end of 2020 at around the same time as the Alpha VOC became widespread. These vaccines also reduced the risk of becoming infected with SARS-COV-2 as well as the risk of transmitting it.[4 69 70] At more recent stages of the pandemic, such as when the Omicron lineage first emerged, the original COVID-19 vaccine from Pfizer (Comirnaty) had become licensed for children ≥5 years old and offered protection from severe disease and at least partial protection from becoming infected and potentially from onward transmission.[71–73] Vaccine uptake, however, remained low for children in most countries. More importantly, a majority of countries with less financial resources and restricted access to health services have not received a sufficient number of vaccine doses to ensure high vaccination coverage even for healthcare workers and vulnerable elderly populations let alone for children and adolescents.[6] Irrespective of the prevalent SARS-CoV-2 lineage this low vaccine uptake and accessibility increased the risk of children and adolescents not only becoming infected themselves but of transmitting the disease to other children and adults. Vaccines for children <5 years old have only recently been licensed in countries with sufficient financial resources.[74 75] In addition to the increased care needs of younger children and thus their higher risk of potential exposure to COVID-19 infected adults, their higher risk of infection and onward transmission compared with older children and adolescents might also be explained by a difference in vaccination status.

## Recommendations for hospitalisation

Recommendations for hospitalisation (as a sign of at least moderate disease) and treatment guidelines for children and adolescents infected with COVID-19 were not standardised, either at the country level or internationally, have changed over the course of the pandemic and might differ between different hospitals within and between countries. In addition, the availability of beds in hospital wards and ICUs varied between countries and over the course of the pandemic greatly influencing the decisions made on hospitalisations. These variations made it impossible to compare the prevalence of moderate and severe COVID-19 disease both between countries or between different pandemic periods in the same country, irrespective of the prevalent SARS-CoV-2 lineage.

## Conclusion

Children are susceptible to infection with a SARS-CoV-2 VOC and can transmit them. Overall patterns of SARS-CoV-2 caused by VOCs among children and adolescents appear similar to those observed in the pre-VOC period. Identification of characteristics associated with specific VOCs or VOC infections in certain age groups is challenged by the frequent lack of data reporting of contextual factors, including the vaccination status of included participants and levels of prevalent SARS-CoV-2. Further studies assessing age-related differences in the risk of infection, the risk of transmission as well as risk factors for severe disease and long-term symptoms or signs after SARS-CoV-2 VOC infection are needed globally. These studies should report the vaccination status of participating adults and children as well as any measures affecting transmission (exposure type and intensity) and hospitalisation (local guidelines) during the study period. The available data are from studies in higher-income settings, where the population is not only assumed to have easier access to health services but where resources for surveillance and research are also more readily available than in low-income and middle-income settings. Owing to a lack of access to vaccines and generally low COVID-19 vaccination coverage, children account for a largely unvaccinated part of the population, especially in low-income and middle-income countries. Local data acquisition and dissemination about SARS-CoV-2 in children and adolescents remain important both for affected countries and worldwide as SARS-CoV-2 continues to mutate and spread.

**Author affiliations**
[1]Medical Service Unit, Swiss Tropical and Public Health Institute, Basel, Switzerland
[2]University of Basel, Basel, Switzerland
[3]Institute of Social and Preventive Medicine, University of Bern, Bern, Switzerland
[4]EHESP French School of Public Health, Rennes, France
[5]Emerging Disease Epidemiology Unit, Insitut Pasteur, Paris, France
[6]Université Paris-Saclay, Gif-sur-Yvette, France
[7]Exposome and Heredity Team, Institut national de la santé et de la recherche médicale, Paris, France
[8]EHESP Rennes-Sorbonne Paris Cité, Rennes, France
[9]Département de pédiatrie, gynécologie et obstétrique, HUG, Geneve, Switzerland
[10]The University of Sheffield, Sheffield, UK

**Acknowledgements** We would like to thank all members of WHO's Maternal, Newborn, Child and Adolescent Health and Ageing department, who contributed to the development of this review: Kathleen Strong, Theresa Diaz, Anshu Banerjee, Maria van Kerkhove, Abdi Mahamud, Valentina Baltag, Judith Ann Mandelbaum-Schmid.

**Contributors** MW, NL and MG conceived and designed the review. MW, LH and MG performed the literature search. MW, AMI, LA-C, NP, YTL, MIA, AGL, IZ, LH and MG screened the articles for relevance and extracted the data. MW, MG and NL adjudicated as third reviewers in case of disagreements during the screening and extraction. MW, AMI, LA-C and MG synthesised the data. MW, AMI, LA-C, LH, NP, YTL, MIA, AGL, NL and MG all discussed the interpretation of the findings. MW and MG wrote the first draft of the manuscript. MI, LA-C, NP, YTL, MIA, AGL, LH and NL all reviewed and revised the manuscript and approved the final version. MW and MG act as guarantors for this study.

**Funding** MW received funding from the WHO (purchase order: 202807927, award grand number: n/a). MIA received the Erasmus Mundus Excellence Scholarship for a European Master of Public Health (2018–2020, award grand number: n/a). He also received subsistence allowance from WHO HQ during an internship and is now receiving a salary from WHO Bangladesh. NL received funds from the Swiss National Science Foundation (project number 176233) and European Union Horizon 2020 research and innovation programme - project EpiPose (grant agreement number 101003688). MG received grants from the Swiss National Science Foundation Ambizione fellowship (PZ00P3_185923) as well as from the Fondation Johanna Dürmüller-Bol (award grand number: n/a).

**Disclaimer** The author is a staff member of the World Health Organization. The author alone is responsible for the views expressed in this publication and they do not necessarily represent the views, decisions or policies of the World Health Organization.

**Map disclaimer** The inclusion of any map (including the depiction of any boundaries therein), or of any geographical or locational reference, does not imply the expression of any opinion whatsoever on the part of BMJ concerning the legal status of any country, territory, jurisdiction or area or of its authorities. Any such expression remains solely that of the relevant source and is not endorsed by BMJ. Maps are provided without any warranty of any kind, either express or implied.

**Competing interests** None declared.

**Patient and public involvement** Patients and/or the public were not involved in the design, or conduct, or reporting, or dissemination plans of this research.

**Patient consent for publication** Not applicable.

**Provenance and peer review** Not commissioned; externally peer reviewed.

**Data availability statement** Data sharing not applicable as no data sets generated and/or analysed for this study.

**Author note** NL and MG are joint last authors.

**ORCID iDs**
Margarethe Wiedenmann http://orcid.org/0000-0002-2357-4133

Lucia Araujo-Chaveron http://orcid.org/0000-0002-2110-6088
Yin Ting Lam http://orcid.org/0000-0002-2380-834X
Muhammad Irfanul Alam http://orcid.org/0000-0002-8759-6585
Nicola Low http://orcid.org/0000-0003-4817-8986
Myrofora Goutaki http://orcid.org/0000-0001-8036-2092

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
