## [Reviewer comments · BMJ Open]

ARTICLE DETAILS

TITLE (PROVISIONAL)	SARS-CoV-2 variants of concern in children and adolescents with COVID-19: a systematic review
AUTHORS	Wiedenmann, Margarethe; Ipekci, Aziz Mert; Araujo-Chaveron, Lucia; Prajapati, Nirmala; Lam, Yin Ting; Alam, Muhammad Irfanul; L'Huillier, Arnaud; Zhelyazkov, Ivan; Heron, Leonie; Low, Nicola; Goutaki, Myrofora

VERSION 1 – REVIEW

REVIEWER	Mona Aldabbagh King Abdulaziz Medical City - Jeddah, Department of Pediatrics
REVIEW RETURNED	08-May-2023

GENERAL COMMENTS	This is a very well-written systematic review that addressed many important questions related to the different childhood SARS CoV2 variants of concern (VOC) and their impact on clinical presentation, severity, risk factors, transmissibility, and outcome. Although there were no language restrictions in the study, however, the study included papers written in languages spoken fluently by the review team only. These languages did not include articles written in Chinese, for which there might be a big weight of papers in Chinese literature considering that it was the area where SARS CoV2 was first reported. It is unclear whether the study included only studies reported during the VOC lineage predominance time or not. If so, then it should be clearly written in the inclusion/exclusion criteria. This unclarity made it confusing to the reader. It was stated in the data synthesis that “The presence of a VOC was determined by genetic sequencing or by assumptions based on authors’ reporting, the dominant lineage at the time of testing, and the study location”, which suggests that the specification of the VOC in the selected articles was not a necessary criterion for inclusion and assumptions were acceptable based on the time and location of the study. Furthermore, in the first paragraph of the discussion section, the authors wrote “Compared to non-VOC and other VOC induced infections more severe infection was observed with either the delta or the gamma SARS-CoV-2 variant”, which suggests the inclusion of both VOC and non-VOC lineages. However, in the search strategy, “VOC” was summed with “COVID-19” with an “AND”. In that case, the VOC should be summed up with COVID-19 in the search strategy by adding “OR” rather than “AND”, because adding “AND” here will exclude many potentially eligible studies that did not indicate the VOC predominating at the time, yet assumptions can be made. Thus, many other potentially eligible papers from the non-VOC era were not included in the review and could not be identified with the used search strategy. Otherwise, if the study is limited to the VOC lineage
--

	predominance periods, then the search strategy is correct. Further suggestions in the search are to include COVID-19 as a MeSH term "COVID-19"[Mesh] and to use the COVID-19 variant of concern MeSH term "SARS-CoV-2 variants" [Supplementary Concept]. Additional comments: -Table 1, second row needs alignment of the items with the listed characteristics
--	---

REVIEWER	David Price University of Melbourne, Centre for Epidemiology & Biostatistics
REVIEW RETURNED	23-Jun-2023

GENERAL COMMENTS	This paper presents a clear summary of SARS-CoV-2 variants in children. Well done to the authors, as there was clearly a lot of work invested to compile these summaries and generate a clear narrative. I have only a couple of minor comments that should be easily addressed:  1. The reference to the earlier systematic review should be included in the Introduction, and more detail on this (and other evidence) provided to justify the role this study plays in filling a gap in the literature. 2. When referring to a 'modelling study', it is important to distinguish between a model-based analysis of original data (i.e., where a mathematical model was fit within a statistical framework to make inference about particular parameters), and a mathematical modelling study that may well not use actual data. I suspect it is the former, but the term 'modelling study' is too vague and should be clarified. 3. In 'Eligibility & Screening', Could the authors clarify "titles and abstracts... a second member verified the selection". Did the second reviewer independently assess all, or only those that the first reviewer had selected? Are the authors comfortable that there were no suitable manuscripts that were not selected by the first reviewer? Not an issue or the data extraction step. 4. In Review Question 2, you report increased odds for severe disease for those infected with Omicron (with an aOR, and 95% CI), but then do not report an equivalent estimate for Delta. It would be helpful to report this estimate as well, so the reader can understand the context for Delta infections as well. 5. Perhaps in the introduction and/or methods, define what age constitutes a child — there are several references of age groups up to 20 years old, though 16+ or 18+ are often considered as adults. This is not necessarily a problem, but what age group you deem as 'children' needs to be clarified. 6. At the first use of non-VOC, please define exactly what this includes, i.e., variants that were not of concern, or only wild-type/ancestral strains? 7. When describing the attack rates of children by age group in Review Question 3, "The attack rates for young children (≤ 4 years old) decreased with increasing age from 44% to 4%.", could the
--

	authors please clarify what the range of age groups were that were considered? I.e., what age did the 4% correspond to? 8. In Review Question 4, bottom page 11, "...daycare setting and found it to be lower (3/24...) than the risk of onward transmission from adults (1/2,...)". I don't think this is a fair statement to make given the sample size, or at least, this statement should be softened to explicitly acknowledge the very limited evidence available. 9. In Review Question 5, emphasis is placed on symptoms >28 days. Presumably this is from symptom onset(?), but my question is just whether there were data on duration of symptoms in the manuscripts reviewed, or if it was only reported as this binary indicator (>28 days, or not)? 10. Regarding recommendations for hospitalisation, it may be worth discussing that it was not just the recommendations for hospitalisation of children that were not standardised and changed over the course of the pandemic, as many countries did not have beds available at different stages of the pandemic either (i.e., the epidemiology potentially played a part too). 11. In "Interpretation and implications...", could the authors please clarify what is meant by 'co-infection' in this case. 12. Very minor note, that figures and tables should be capitalised throughout (e.g., Table 1, Supplementary Table 2) as should variants of concern (e.g., Omicron). In Results, "non was from a lower..." should be "none were...". In the last line of "Risk of exposure to...", you can remove ", however, also" such that this line reads "these findings could be partly explained by...".
--	--

VERSION 1 – AUTHOR RESPONSE

Reviewer: 1

Dr. Mona Aldabbagh, King Abdulaziz Medical City - Jeddah Comments to the Author:

This is a very well-written systematic review that addressed many important questions related to the different childhood SARS CoV2 variants of concern (VOC) and their impact on clinical presentation, severity, risk factors, transmissibility, and outcome.

Although there were no language restrictions in the study, however, the study included papers written in languages spoken fluently by the review team only. These languages did not include articles written in Chinese, for which there might be a big weight of papers in Chinese literature considering that it was the area where SARS CoV2 was first reported.

We would like to thank the reviewer for this positive feedback. We appreciate her comment about the coverage of our search strategy and acknowledge this in the 'Discussion – Strengths and limitations' (page 13), "Our search strategy might have missed relevant studies, although we only excluded three records for language reason." The emergence of

SARS-CoV-2 in China is more relevant for studies performed at the beginning of the pandemic. Our review covers a later period, after VOCs started to circulate.

It is unclear whether the study included only studies reported during the VOC lineage predominance time or not. If so, then it should be clearly written in the inclusion/exclusion criteria. This unclarity made it confusing to the reader. It was stated in the data synthesis that "The presence of a VOC was determined by genetic sequencing or by assumptions based on authors' reporting, the dominant lineage at the time of testing, and the study location", which suggests that the specification of the

VOC in the selected articles was not a necessary criterion for inclusion and assumptions were acceptable based on the time and location of the study.

We apologise for the lack of clarity. We have added a clarification and moved the determination of the circulating VOC to Methods – Eligibility and screening (page 5, paragraph 4).

Observational epidemiological studies... were eligible for inclusion if data collection occurred during a period when VOCs were predominant in the study setting. The presence of a VOC could be based on genetic sequencing in the study or by assumptions based on the authors' reporting, the dominant lineage at the time of testing, and the study location.

Furthermore, in the first paragraph of the discussion section, the authors wrote "Compared to non-VOC and other VOC induced infections more severe infection was observed with either the delta or the gamma SARS-CoV-2 variant", which suggests the inclusion of both VOC and non-VOC lineages. However, in the search strategy, "VOC" was summed with "COVID-19" with an "AND". In that case, the VOC should be summed up with COVID-19 in the search strategy by adding "OR" rather than "AND", because adding "AND" here will exclude many potentially eligible studies that did not indicate the VOC predominating at the time, yet assumptions can be made. Thus, many other potentially eligible papers from the non-VOC era were not included in the review and could not be identified with the used search strategy. Otherwise, if the study is limited to the VOC lineage predominance periods, then the search strategy is correct.

The reviewer is correct; our review only included studies performed when VOC lineages were predominant so the search strategy is correct. All of our five review questions focused on VOC specific infections as described in the abstract (page 2).

We clarified this further in the manuscript:

Data synthesis (page 7, paragraph 1)

Any comparisons with non-VOC lineages (any strain not considered by WHO as a variant of concern) included in this manuscript were made by the authors of the original manuscripts.² We did not make independent comparisons between outcomes during VOC and non-VOC periods.

Further suggestions in the search are to include COVID-19 as a MeSH term "COVID- 19"[Mesh] and to use the COVID-19 variant of concern MeSH term "SARS-CoV-2 variants" [Supplementary Concept].

Thank you for the suggestion. We considered this while building our search strategy and we tested and compared the results between different combinations including our original

search terms and search terms including the above-mentioned MeSH terms. We found that using the MeSH terms significantly broadened our search results but did not identify additional relevant studies. We have therefore retained our chosen search strings, as described in the 'Methods - Search Strategy' (page 5, paragraph 3).

Additional comments:

-Table 1, second row needs alignment of the items with the listed characteristics Thank you, we corrected this.

Reviewer: 2

Dr. David Price, University of Melbourne, Doherty Institute Comments to the Author:

This paper presents a clear summary of SARS-CoV-2 variants in children. Well done to the authors, as there was clearly a lot of work invested to compile these summaries and generate a clear narrative. I have only a couple of minor comments that should be easily addressed:

Thank you for this positive feedback.

1. The reference to the earlier systematic review should be included in the Introduction, and more detail on this (and other evidence) provided to justify the role this study plays in filling a gap in the literature.

We now refer to the earlier systematic review, which covered the period before the circulation of VOCs in the 'Introduction'. We aimed to introduce evidence and reasons for updating our review in paragraphs 1 and 2 on page 4. We have revised the following paragraphs as follows ('Introduction', page 4, paragraph 1):

"In our systematic review, covering the period when SARS-CoV-2 ancestral strains circulated, we found that children (≤ 12 years of age) and adolescents (13–17 years of age) usually present with mild disease, with a minority of children of all ages (< 18 years) remaining asymptomatic throughout the course of infection, and their risks of acquiring and transmitting SARS-CoV-2 seemingly increases with age.¹"

And ('Introduction', page 4, paragraph 2):

A re-evaluation of the role of children and adolescents in the SARS-CoV-2 pandemic is warranted, given the potential for the mutations in VOCs to affect age-associated patterns of COVID-19 clinical disease and SARS-CoV-2 transmissibility, and the higher numbers of children and adolescents testing positive for SARS-CoV-2 than in the pre-VOC period.

2. When referring to a 'modelling study', it is important to distinguish between a model-based analysis of original data (i.e., where a mathematical model was fit within a statistical framework to make inference about particular parameters), and a mathematical modelling study that may well not use actual data. I suspect it is the former, but the term 'modelling study' is too vague and should be clarified.

The reviewer is correct. We clarified this in the 'Methods – Eligibility and Screening' (page 5, paragraph 4):

Observational epidemiological studies (case series with more than one case, cross-sectional studies, case-control studies and cohort studies) and modelling studies that were based on

analysis of the original data were eligible for inclusion if data collection occurred during a period when VOCs were predominant in the study setting.

3. In 'Eligibility & Screening', Could the authors clarify "titles and abstracts... a second member verified the selection". Did the second reviewer independently assess all, or only those that the first reviewer had selected? Are the authors comfortable that there were no suitable manuscripts that were not selected by the first reviewer? Not an issue or the data extraction step.

Thank you for the opportunity to clarify the screening process. We did not conduct independent duplicate screening. Although we might have excluded eligible manuscripts, we took an inclusive approach.

We have adjusted the manuscript accordingly ('Methods – Eligibility and Screening', page 6, paragraph 1).

Identified titles and abstracts were screened for potential eligibility by one member from an international team of ten reviewers, including records about which they were unsure. A second member verified the selection. If either reviewer was unsure about the potential eligibility, we obtained the full-text. We followed the same approach for the full-text assessment.

4. In Review Question 2, you report increased odds for severe disease for those infected with Omicron (with an aOR, and 95% CI), but then do not report an equivalent estimate for Delta. It would be helpful to report this estimate as well, so the reader can understand the context for Delta infections as well.

We added the equivalent estimate for severe disease for those infected with Delta ('Results', page 10, paragraph 1):

...if they were infected with the Omicron lineage (aOR 95%CI 3.2, 1.1-9.0), but not if they were infected with the Delta lineage (aOR 95%CI 0.8, 0.5-1.4).

5. Perhaps in the introduction and/or methods, define what age constitutes a child — there are several references of age groups up to 20 years old, though 16+ or 18+ are often considered as adults. This is not necessarily a problem, but what age group you deem as 'children' needs to be clarified.

We agree with the need for definition and considered all those younger than 20 years old, although definitions differed between studies. In the 'Supplementary Tables 7-12', we provide the age ranges for each study if they were available. We clarified this in the 'Introduction' (page 4, paragraph 3):

The objectives of this study were to address five research topics about SARS-CoV-2 VOC infections in children and adolescents younger than 20 years:

6. At the first use of non-VOC, please define exactly what this includes, i.e., variants that were not of concern, or only wild-type/ancestral strains?

Depending on the study it might include both. We clarified this in the 'Methods – Data synthesis' (page 7, paragraph 1):

Any comparisons with non-VOC lineages (any strain not considered by the World Health Organization as a variant of concern) included in this manuscript were made by the authors of the included manuscripts.²

7. When describing the attack rates of children by age group in Review Question 3, "The attack rates for young children (≤ 4 years old) decreased with increasing age from 44% to 4%.", could the authors please clarify what the range of age groups were that were considered? I.e., what age did the 4% correspond to?

The article we refer to (ref 14) included children from 0-4 and the lowest (4%) attack rate corresponded to the older age group (3-4 year olds). We revised the manuscript to clarify this ('Results', page 10, paragraph 4):

The attack rates for young children (≤ 4 years old) decreased with increasing age from 44% to 4%, with the highest rate for infants < 1 year old and the lowest for children 3-4 years old.

8. In Review Question 4, bottom page 11, "...daycare setting and found it to be lower (3/24...) than the risk of onward transmission from adults (1/2,...)". I don't think this is a fair statement to make given the sample size, or at least, this statement should be softened to explicitly acknowledge the very limited evidence available.

We agree with the reviewer's remark. We revised this section as follows to explicitly acknowledge the limited evidence availability ('Results', page 12, paragraph 1):

The availability of evidence on the risk of onward transmission of children with an Omicron variant infection was limited. We only found one small study that analysed the risk of onward transmission of young children ≤ 6 years old in a daycare setting and found it to be lower (3/24, 12.5%) than the risk of onward transmission from adults (1/2, 50%).

9. In Review Question 5, emphasis is placed on symptoms > 28 days. Presumably this is from symptom onset(?), but my question is just whether there were data on duration of symptoms in the manuscripts reviewed, or if it was only reported as this binary indicator (> 28 days, or not)?

We did not extract and synthesise information on the duration of symptoms from the studies included in this review. The > 28 days/4 weeks cut-off from symptom onset was predefined by the US Department of Health and Human Services as part of the definition for the first symptoms associated with long-COVID.^[5] Therefore, we included studies that referred specifically to long COVID or to a symptom duration longer than 28 days to answer this review question. Overall, the duration of symptoms was rarely reported in most studies.

We clarified this in the 'Introduction' (page 5, paragraph 1):

1. Does the risk of developing long-term effects (duration >28 days) after infection with a SARS-CoV-2 VOC differ between children, adolescents and adults who have not received a COVID-19 vaccine?

and in the 'Results' (page 12, paragraph 2):

Review question 5: Risk of long-term symptoms or signs (duration >28 days) following an infection with a SARS-CoV-2 VOC

Few children reported long-term symptoms or signs of infection with a defined duration of >28 days.^{32,33}

Reference(s):

5. Department of Health and Human Services. What is Long COVID? [Available from: <https://www.covid.gov/longcovid/definitions.>]

10. Regarding recommendations for hospitalisation, it may be worth discussing that it was not just the recommendations for hospitalisation of children that were not standardised and changed over the course of the pandemic, as many countries did not have beds available at different stages of the pandemic either (i.e., the epidemiology potentially played a part too).

We want to thank the reviewer for raising this point of available beds on wards and ICUs and how it varied at different stages of the pandemic. We included this point in our 'Discussion' (pages 15-16, paragraph 2/1):

Recommendations for hospitalisation (as a sign of at least moderate disease) and treatment guidelines for children and adolescents infected with COVID-19 were not standardised, either at the country level or internationally, have changed over the course of the pandemic and might differ between different hospitals within and between countries. In addition, the availability of beds on wards and ICUs varied between countries and over the course of the pandemic, which would influence the data about hospitalisation.

11. In "Interpretation and implications...", could the authors please clarify what is meant by 'co-infection' in this case.

We regard co-infections as infections with other respiratory viruses. We have adapted the manuscript to include this information as follows ('Discussion', page 14, paragraph 1)

... and different patterns of comorbidity and co-infection with other respiratory viruses.

12. Very minor note, that figures and tables should be capitalised throughout (e.g., Table 1, Supplementary Table 2) as should variants of concern (e.g., Omicron). In Results, "non was from a lower..." should be "none were...". In the last line of "Risk of exposure to...", you can remove " , however, also" such that this line reads "these findings could be partly explained by...".

Thank you for raising these points. We have included this feedback in the manuscript.

VERSION 2 – REVIEW

REVIEWER	Mona Aldabbagh King Abdulaziz Medical City - Jeddah, Department of Pediatrics
REVIEW RETURNED	03-Sep-2023
GENERAL COMMENTS	Thank you respected author for the great effort in summarizing such huge data systematically. I have nothing to add.

REVIEWER	David Price University of Melbourne, Centre for Epidemiology & Biostatistics
REVIEW RETURNED	29-Aug-2023
GENERAL COMMENTS	Thank you to the authors for the revised manuscript, all of my queries have been addressed.